

# A deep descriptor for cross-tasking EEG-based recognition

Mariana R.F. Mota[1], Pedro H.L. Silva[1], Eduardo J.S. Luz[1], Gladston J.P. Moreira[1], Thiago Schons[1], Lauro A.G. Moraes[1] and David Menotti[2]

[1] Department of Computing, Universidade Federal de Ouro Preto, Ouro Preto, Minas Gerais, Brazil
[2] Department of Informatics, Universidade Federal do Paraná, Curitiba, Paraná, Brazil

## ABSTRACT

Due to the application of vital signs in expert systems, new approaches have emerged, and vital signals have been gaining space in biometrics. One of these signals is the electroencephalogram (EEG). The motor task in which a subject is doing, or even thinking, influences the pattern of brain waves and disturb the signal acquired. In this work, biometrics with the EEG signal from a cross-task perspective are explored. Based on deep convolutional networks (CNN) and Squeeze-and-Excitation Blocks, a novel method is developed to produce a deep EEG signal descriptor to assess the impact of the motor task in EEG signal on biometric verification. The Physionet EEG Motor Movement/Imagery Dataset is used here for method evaluation, which has 64 EEG channels from 109 subjects performing different tasks. Since the volume of data provided by the dataset is not large enough to effectively train a Deep CNN model, it is also proposed a data augmentation technique to achieve better performance. An evaluation protocol is proposed to assess the robustness regarding the number of EEG channels and also to enforce train and test sets without individual overlapping. A new state-of-the-art result is achieved for the cross-task scenario (EER of 0.1%) and the Squeeze-and-Excitation based networks overcome the simple CNN architecture in three out of four cross-individual scenarios.

## INTRODUCTION

Nowadays, different biometric modalities are being explored. They have been gradually replacing the logging/password systems, once it represents the future path in terms of digital security. Among them, one of the categories which has gained attention is the vital signals based biometric modalities, such as the electrocardiogram (ECG) (*Luz et al., 2018*; *Silva et al., 2019*; *Garcia et al., 2017*) and the electroencephalogram (EEG) (*Schons et al., 2017*; *Poulos et al., 1999*; *Carrión-Ojeda, Fonseca-Delgado & Pineda, 2020*). The EEG, initially used for clinical purposes, have already been used by consumers and incorporated into portable devices for applications not directly related to medicine, as in the gaming industry, education systems aimed at monitoring engagement and in EEG-based brain-computer interface (*Sawangjai et al., 2019*). Nonetheless, automatic disease detection (*Islam, Rastegarnia & Yang, 2015*) and an individual identification employing brain waves are still challenging tasks (*Yang & Deravi, 2017*).

Corresponding author
Gladston J.P. Moreira,
gladston@ufop.edu.br

The EEG signal is the measurement of the electrical activity of the brain captured by electrodes strategically positioned on the scalp (*Boubakeur et al., 2017*). The first use of the EEG signal as biometrics dates back to 1980 in Stassen's seminal work (*Stassen, 1980*), using a speech recognition technique to characterize the EEG spectrum pattern of a person. Since then, several approaches have investigated the EEG as a biometric modality (*Das, Maiorana & Campisi, 2018*; *Fraschini et al., 2015*; *Yang, Deravi & Hoque, 2018*; *Gui et al., 2019*).

Researches on EEG as biometrics have shown that this signal satisfies three out four requirements to be considered as a biometric modality (*Jain, Ross & Prabhakar, 2004*): (1) collectability: it can be measured and quantified; (2) unicity: it provides distinction among individuals; (3) universality: the element exists in all person. The fourth requirement, persistence over time, is still an open problem in the literature (*Marcel & Millán, 2007*; *Kostílek & Šťastny, 2012*).

Many authors have addressed the EEG signal as a biometric modality with traditional machine learning techniques (*Fraschini et al., 2015*; *Singh, Mishra & Tiwary, 2015*; *Yang, Deravi & Hoque, 2018*). Today, deep learning represents the state-of-the-art for several computer vision and pattern recognition problems and also there are works in the literature addressing deep-learning-based biometrics on EEG (*Ma et al., 2015*; *Das, Maiorana & Campisi, 2017*, *2018*; *Mao, Yao & Huang, 2017*; *Wang et al., 2019*; *Wilaiprasitporn et al., 2019*). Although it is well known that methods based on deep learning need large amounts of data, especially, deep convolution-based architectures, a workaround for this issue is to artificially create new samples, such as the data augmentation techniques. In a preliminary work (*Schons et al., 2017*), a data augmentation technique was proposed to increase the training data, improving the network performance, and enabling the deep learning model to converge for the Physionet—EEG Motor Movement/Imagery Dataset.

It has been shown in previous works evidence that EEG has biometric potential under the performance of different tasks (*Vinothkumar et al., 2018*, *DelPozo-Banos et al., 2018*; *Kong et al., 2018*; *Kumar et al., 2019*; *Fraschini et al., 2019*). In this work, the explore EEG as a biometric modality under this challenging multi-task perspective. In a real situation, a person can be in motion, or even, imagining an action, and with that, generating interference that can be captured by the EEG signal. Thus, a deep feature descriptor approach is proposed in order to mitigate the impact of movement/imagery tasks on the EEG biometrics. A evaluation whether EEG biometrics benefit from any specific task, or from its nature (motor or imaginary) is conducted. The relevance of the number of channels is also investigated as well as a more robust model based on Squeeze-and-Excitation blocks (*Hu, Shen & Sun, 2018*). The Squeeze-and-Excitation blocks promote a channel-wise feature response and since the EEG signal can have up to 64 channels and each channel captures different nuances of brain movement, these blocks are a promising research path for this type of biometric modality.

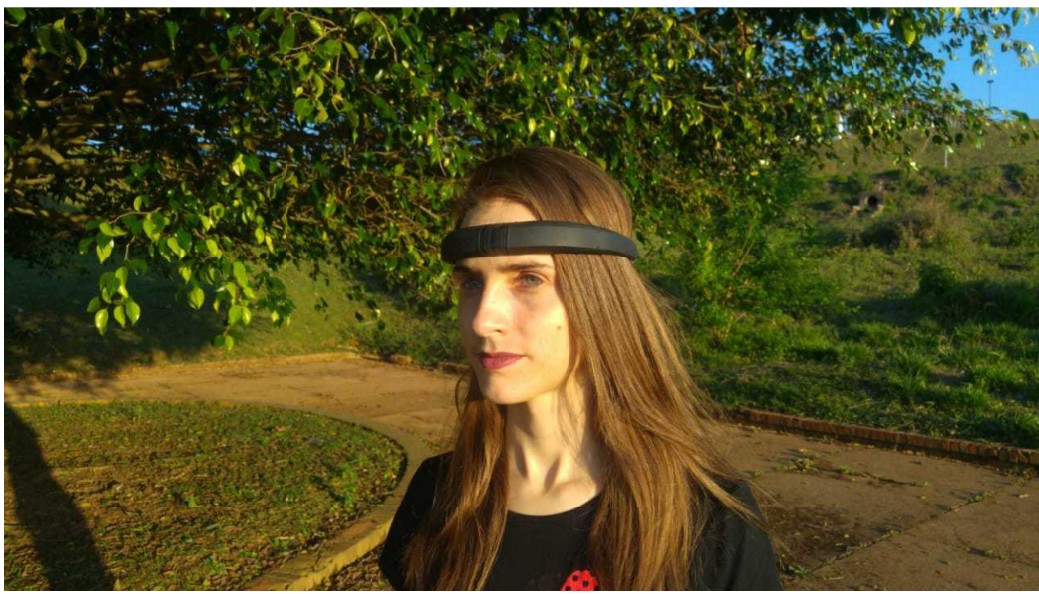

**Figure 1 Example of an EEG headband with few electrodes and no need for conductor gel.**

This work extends the one presented in the 22nd CIARP (Iberoamerican Congress on Pattern Recognition) (*Schons et al., 2017*). Here, the following improvements are described:

- It presents results for a cross-task (one task is used to train and another to test) multi-task (two or more tasks are used to train and another to test), and cross-individual scenario (the individual used to train is not used to test) extending the approach showed in *Schons et al. (2017)*, considering motor task interference.
- It presents an extensive literature review.
- It explores the Squeeze-and-Excitation blocks instead of conventional CNN blocks.
- A better description of the methodology/experimentation process and detailed analysis regarding parameter tuning.

The contributions of this work are summarized as follow:

- The new state-of-the-art method for EEG-based biometric verification evaluated in a protocol aiming a cross-task scenario.
- A data augmentation technique based on the exploration of the overlap between signals used during the training phase.
- Competitive results with fewer EEG channels.
- An in-depth discussion about the implications of using multiple motors/imaginary tasks for biometrics with EEG.

The results reported in this work corroborates with the feasibility of the EEG as a biometric modality, once the evaluation carried with different tasks in different runs have

reported promising results. Although our experiments reveal that physical motor tasks (not imaginary) hinder the task of identification, making the problem more challenging.

With more ubiquitous acquisition method (as illustrated in Fig. 1) and the advancement of hardware accelerators for deep learning, there is a large potential for its usage in real-life applications.

The remaining of this work is organized as follows. Related works are presented in the second section. In the third section, the proposed methodology is presented, while the experiments and results are described in the fourth section. Finally, the conclusions are pointed out in the fifth section.

## DATASET AND RELATED WORKS

In this section, a popular benchmark dataset is presented: the EEG Motor Movement/ Imagery dataset (*Fraschini et al., 2015*; *Yang, Deravi & Hoque, 2018*; *Sun, Lo & Lo, 2019*), available in Physionet Resource (https://physionet.org/content/eegmmidb/1.0.0/). The EEG Motor Movement/Imagery dataset is one of the most comprehensive datasets of EEG signals and also contemplates motor tasks during the acquisition sessions.

The majority of the works presented in this section use the EEG Motor Movement/ Imagery dataset (*Goldberger et al., 2000*; *Schalk et al., 2004*). However, other datasets were considered by authors in the literature; thus, the presented methods are categorized according to the signals type used in the evaluation: baseline task or single-task and multi-task.

### EEG motor movement/imagery dataset

The EEG Motor Movement/Imagery dataset is a popular open-access dataset that consists of over 1,500 recordings of EEG signals obtained from 109 volunteers, with one-minute or two-minutes lengths. The EEG signal was obtained from 64 electrodes distributed over the scalp. The EEG subset contains a total of 109 subjects and 14 acquisition records. From the acquisition records, there are two motor and two imaginary task types and two baseline tasks.

The baseline tasks capturing occurred in a controlled scenario in which the individual follows the command to open and close their eyes during specifics moments. Except for the baseline, all other tasks have three acquisition sessions.

The records of three sessions of four different motor/imaginary tasks result in 12 sessions. Regarding the 12 sessions, there are four distinct motor or imaginary task types (T1–T4), each one with three runs (R1–R3). The activities assigned to the tasks are:

- Task 1 (T1): In a screen, a target appears on the left or right side, and a subject is requested to open and close the corresponding fist on the same side until the target disappears. After that, the subject may relax.
- Task 2 (T2): In a screen, a target appears on the left or right side, and the subject is requested to imagine opening and closing the corresponding fist on the same side until the target disappears. After that, the subject may relax.

- Task 3 (T3): In a screen, a target appears at the top or bottom of a screen. If the target is on the top, the subject is asked to open and close both fists. Otherwise, if the target is in on the bottom, the subject is requested to open and close both feet. This task runs until the target disappears. After that, the subject may relax.

- Task 4 (T4): On a screen, a target appears at the top or bottom of a screen. If the target is on the top, the subject is asked to imagine opening and closing both fists. Otherwise, if the target is in on the bottom, the subject is requested to imagine opening and closing both feet. This task runs until the target disappears. After that, the subject may relax.

## Related works

It is possible to summarize the related works evaluated under the EEG Motor Movement/Imagery dataset into two groups: (1) the ones focused on the baseline task, i.e., eyes close (EC) and eyes open (EO) tasks, and (2) the ones focused on the multi-task recognition. Regarding the first group, the individuals are in a rest state with the eyes closed or opened (*Del Pozo-Banos et al., 2014*). In contrast, in the second group, the individuals were requested to do simple motor or imaginary tasks.

The work in *Gui et al. (2019)* presents a survey on EEG signals biometry. The authors reviewed the state-of-the-art for the recognition task and concluded that the EEG may has some advantages over other biometrics markers. One of the significant advantages of the EEG to consider is that the user must agree and consent to record the EEG signal what difficulties the theft of this kind of signal. *Gui et al. (2019)* also explored all steps regarding the EEG signal, from the acquisition to the classification, and described several EEG datasets from the literature. Moreover, the authors also detected open problems related to brain biometrics, such as multi-modality and the persistence of the EEG. According to the authors, the EEG signal has a great potential as biometry from the theoretical point of view and due to results obtained from empirical experiments.

*Singh, Mishra & Tiwary (2015)* proposed an approach to find discriminative characteristics described by unique patterns from the relations generated among the cerebral regions. The authors preprocessed the data by applying a low-pass filter to reduce the noise and used a technique called Magnitude Squared Coherence, which relies on the phase constancy while a brain area is interacting with another, generate the signal descriptors. The classification process performed the K-Nearest Neighbors (KNN) algorithm. The proposed approach evaluated 64, 10, 6, and 5 channels for the EEG signal capturing and achieved a final accuracy score for the identification reported task of 100% with 64 electrodes. This experiment corroborates the hypothesis that the more channels/information employed, the better will be the result.

*Fraschini et al. (2015)* evaluated a biometric verification approach using all the 64 electrodes available in the dataset. The proposed approach can be divided into four steps: (1) filter the raw data to allow a study of the specific frequency data from the cerebral network; (2) estimate the statistical independency in pairs among EEG temporal series; (3) create a weighted graph where each edge represents a functional connection among the

connected electrodes on the head; and (4) perform a characterization of the cerebral functional organization aiming a centrality measure to qualify the importance of each node inserted in the graph. The result is a 64-length-vector used to classify and to report the results segmented by the frequency. The authors reported an EER of 4.4% using the gamma frequency (30–50 Hz) and observed a difficulty enhancement for individual verification when the frequency band gets slower.

CNNs are also evaluated for the EEG problem. *Ma et al. (2015)* created a CNN architecture with two convolution layers, two pooling layers, and one fully connected layer. However, the authors did not use all 109 individuals. Instead, *Ma et al. (2015)* evaluated their approach with 10 individuals for testing. The authors divided the 60 s of EO and EC in 55 s in a one-second fragment for training and the remaining 5 s for testing. The identification rate reported was in the range of 64% to 86%.

*Das, Maiorana & Campisi (2017)* evaluated a CNN with four convolutional layers, two max-pooling, one rectified linear unit (ReLU), and a softmax-loss layer. The dataset used to perform the experiments were acquired from 40 subjects over two distinct sessions separated by a week in a visual stimuli environment. The correct recognition rate (CRR) reported was in the range between 80.65% and 98.8% in the best scenario.

*Das, Maiorana & Campisi (2018)* also performed EEG biometry with a CNN containing four convolutional layers, two max-pooling, a ReLU, and a softmax-loss layer. The dataset evaluated consists of 40 subjects performing imaginary arms and legs movements collected in two sessions with an interval of 2 weeks. The authors achieved the accuracy scores of 81.25% and 93% for rank-1 and rank-2, respectively.

*Mao, Yao & Huang (2017)* proposed another approach using CNN. The authors used convolutional layers with ReLu and max-polling, followed by fully connected layers with softmax. The evaluation was made in a dataset containing data of 100 subjects using 64 channels during a driving experiment. The CRR achieved was 97%.

*Maiorana & Campisi (2018)* analyzed the discriminative characteristics of EEG in longitudinal behavior, aiming to verify the pertinence across time. The study utilized the signals captured from 45 users during approximately 36 weeks from the first EEG data recording in five to six sessions. The results showed that aging could damage the EEG traits, but yet the authors could achieve EER below 2% from one data collection to another distant in time.

*Wang et al. (2019)* evaluated the impact of different motor and imaginary tasks in the identification scenario. The authors used all the 109 individuals of the EEG Motor Movement/Imagery Dataset and also created a dataset with 59 subjects performing attention and descriptive tasks. The proposed approach is a Graph Convolutional Neural Network-based to extract discriminating features from the EEG graphs created by the Phase Locking Value (PLV) algorithm. Using the same task to train and test, the authors reported an average CRR of 99.96% and 98.94% for the EEG Motor Movement/Imagery Dataset and the created datasets, respectively. A degradation of 86.21% was observed when trained on resting-state data and tested in diverse states in the EEG Motor Movement/Imagery Dataset. However, mixing signals in different states for training the

GCNN resulted in an average CRR of 99.98% with the Physionet bank and 99.96% with the author's personal acquisition bank.

*Yang, Deravi & Hoque (2018)* also evaluated the impact of motor/imaginary tasks during the recognition process in biometric identification and verification tasks. The authors used a discrete Wavelet transform in each electrode and created the feature vector as the concatenation of the standard deviations of all electrodes. The vector created is used as the input for a Linear Discriminant Analysis (LDA) classifier. A majority voting rule merges all classifier decisions of all time windows. In one of the experiments performed, the authors trained on tasks T1 + T2, tested on T1R2 using 9 electrodes and reached an identification accuracy of 100% and an EER of 2.63%. They also analyzed the impact of using only 3 channels and achieved a maximum accuracy of 89% when training with T1R1 + T1R3 and testing with T1R2. In the verification scenario, the EER is approximately three times greater using the reduced number of electrodes.

To find the best set of electrodes for an identification problem, *Alyasseri et al. (2020)* used the EEG Motor Movement/Imagery Dataset as a single matrix containing the signals from all tasks performed by the 109 individuals. The authors propose a hybrid optimization technique to find the most relevant channels to extract discriminant characteristics, using the Flower Pollination Algorithm (FPA) and the $\beta$-hill-climbing algorithm ($\beta$-hc). Besides, the proposed approach presented a study of the best domain to extract the characteristics of an EEG signal, comparing the use of characteristics in the time domain, frequency domain, and both simultaneously. The hybrid method FPA$\beta$-hc obtained better results than isolated FPA in most cases. The accuracy reported of 96.05% was achieved using a Support Vector Machine (SVM) classifier with the RBF kernel and the characteristics in both the time and frequency domain simultaneously.

In the work proposed in *Sun, Lo & Lo (2019)*, the authors applied a hybrid convolution and recurrent deep neural network with a Long-Short Time Memory (LSTM) for individual identification. The authors tested the use of 4, 16, 32, and 64 electrodes. Besides, instead of using 12-seconds length segments, a 1-second length segment was used and reached 0.41% of EER with 16 electrodes. The focus of their discussion is the trade-off between performance and the use of the electrodes/recording time. The authors outweighed the loss in EER to the reduction of the electrodes, and, according to the authors, 12-second segments are not feasible for real applications. Despite the outstanding results presented by the authors, it is worth highlighting that the protocol is different from the one showed in this work and in *Yang, Deravi & Hoque (2018)*. The authors evaluated two different scenarios compared to the one presented in *Yang, Deravi & Hoque (2018)*. The rationale for the first scenario is that the authors use 90% to train/validate and 10% to test, that is, the same individual signal is present in training and testing. This scenario is more straightforward than the one proposed by *Yang, Deravi & Hoque (2018)*. The reported results cannot be compared to the presented in *Yang, Deravi & Hoque (2018)*, once it uses only a one-second signal, and the evaluation protocol is different.

Some works in the literature do not use the EEG Motor Movement/Imagery dataset but present relevant results and alternative techniques that inspired the development of this work.

The work of *El-Fiqi et al. (2018)* proposed a CNN with raw steady-state visual evoked potentials (SSVEPs) for individual identification and verification. In the SSVEPs acquisition protocols, the subject focuses its attention on a repetitive visual stimulus while recording the activity response signals. The study explores two SSVEP datasets with raw data of four and ten subjects that were seated and focused on three groups of flashing LED stimuli blinking at 13 Hz, 17 Hz, and 21 Hz frequencies. The signals were recorded at 256 Hz by eight electrodes placed on the parietal-occipital area. The performance of three classical methods (support vector machine, random forest, and shallow feed-forward network with one layer) with spectral features were compared against the proposed CNN method, which works directly with the raw signal. The deep learning-based approach achieved an averaged identification accuracy of 96.80% and an averaged verification accuracy of 98.34%. These results outperform those obtained by other classical methods. Besides, the proposed CNN needs no complex techniques for feature representation or extraction to achieve good results, making it feasible for real-time systems.

Table 1 summarizes these related works.

## PROPOSED APPROACH AND EVALUATION SCENARIOS

In this section, the proposed method is presented and outlined as the pipeline shown in Fig. 2. The method is evaluated on the EEG Motor Movement/Imagery Dataset (*Goldberger et al., 2000*; *Schalk et al., 2004*), under the three scenarios evaluated in this work and presented in "Results and Discussion".

In order to execute the pipeline proposed in Fig. 2, preprocessing methodology is developed along with the data augmentation protocol, the CNN model, and the respective data representation along with the evaluation process. The steps presented in Fig. 2 are described in details in the next subsections.

### Data preprocessing

Biometrics, disease detection, and brain death detection are two of many applications that could be employed based on the EEG signal. Regardless of the application, preprocessing is required since the signal is very susceptible to noise. Several approaches have been used in the literature for data processing, such as Common Spatial Pattern (CSP) (*Yong, Ward & Birch, 2008*), Wavelet transforms (*Kumar et al., 2009*; *Ghandeharion & Ahmadi-Noubari, 2009*), and Independent Component Analysis (ICA) (*Delorme, Sejnowski & Makeig, 2007*).

For biometrics purposes, the band-pass FIR/IIR filters are popular preprocessing techniques. Once each frequency is related to a specific cerebral activity (*Boubakeur et al., 2017*), filtering out unwanted frequency-bands aids in the detection process. In this work, a band-pass filter is applied aiming only the intermediate frequency-bands of the signal spectrum. Since the data acquired by equipment used in Physionet EEG Motor Movement/Imagery Dataset was sampled at 160 Hz, it has well-defined components up to 80 Hz. Besides, it is not usual to employ the full spectrum, since some frequency bands are more discriminant than others for biometry (*Fraschini et al., 2015*). According to *Yang & Deravi (2017)*, signals related to EEG biometrics have higher energy in the

**Table 1 Related works for EEG-based biometric.**

| Work | Database | Classes | Acquisition | Approach | Channels | Result |
|------|----------|---------|-------------|----------|----------|--------|
| *Singh, Mishra & Tiwary (2015)* | Physionet | 109 | Motor/Imaginary tasks | Magnitude Squared Coherence | 64 | Acc = 100% |
| *Fraschini et al. (2015)* | Physionet | 109 | Rest | Eigenvector | 64 | EER = 4.4% |
| *Ma et al. (2015)* | Physionet | 10 | Rest | CNN | 64 | Acc = 88% |
| *Das, Maiorana & Campisi (2017)* | Own | 40 | Visual stimuli | CNN | 17 | CRR = 98.8% |
| *Das, Maiorana & Campisi (2018)* | Own | 40 | Imaginary arms/legs movement | CNN | 17 | CRR = 93% |
| *Mao, Yao & Huang (2017)* | BCIT | 100 | Driving car | CNN | 64 | CRR = 97% |
| *Maiorana & Campisi (2018)* | Own | 45 | Sit | Hidden Markov models (HMMs) | 9 | EER < 2% |
| *Wang et al. (2019)* | Physionet | 109 | Motor/Imaginary tasks | GCNN + PLV | 64 | CRR = 99.98% FAR = 1.65% |
| *Yang, Deravi & Hoque (2018)* | Physionet | 108 | Motor/Imaginary tasks | Discrete Wavelet Transform + LDA | 9 | Acc = 100% EER = 2.63% |
| *Alyasseri et al. (2020)* | Physionet | 109 | Motor/Imaginary tasks | FPA + $\beta$-hill | 35 | Acc = 96.05% |
| *Sun, Lo & Lo (2019)* | Physionet | 109 | Motor/Imaginary tasks | 1D-Conv. LSTM | 16 | EER = 0.41% |
| *El-Fiqi et al. (2018)* | SSVEP database | 14 | Visual stimulus | CNN | 8 | Verification Acc = 98.34% |

frequency spectrum below 50 Hz. In that manner, filters are built for the following bands: 01–50 Hz, 10–30 Hz, and the gamma band (30–50 Hz), following the ones proposed by *Fraschini et al. (2015)*.

The data division in training and testing dataset is defined according to the evaluated scenario. For the first one (baseline task evaluation), the EC is used for testing. For the second scenario (multi-task evaluation), the data division proposed by *Yang & Deravi (2017)* is followed. In the third scenario (cross-task evaluation), proposed in this work, the T1R2 is used as testing. Finally, the forth scenario (cross-individual evaluation), also proposed in this work, the EC and EO are used for testing.

## Data augmentation

The evaluation protocol in *Fraschini et al. (2015)* proposes the division of the EEG data into 12-second segments (1,920 samples). Since the recording motor/imagery tasks have 120 s, each task has a total of 10 segments of 12 s without overlap. From both 60-second baseline recording sessions (EO and EC), five segments are created, disregarding the overlap. A total of only five or ten segments for each individual is not enough for the CNN training to converge.

A workaround for this issue is a data augmentation technique for the training set. The strategy consists of abundantly extract segments with overlapping from the EEG segments. In this work, the impact of the data augmentation technique based on segments

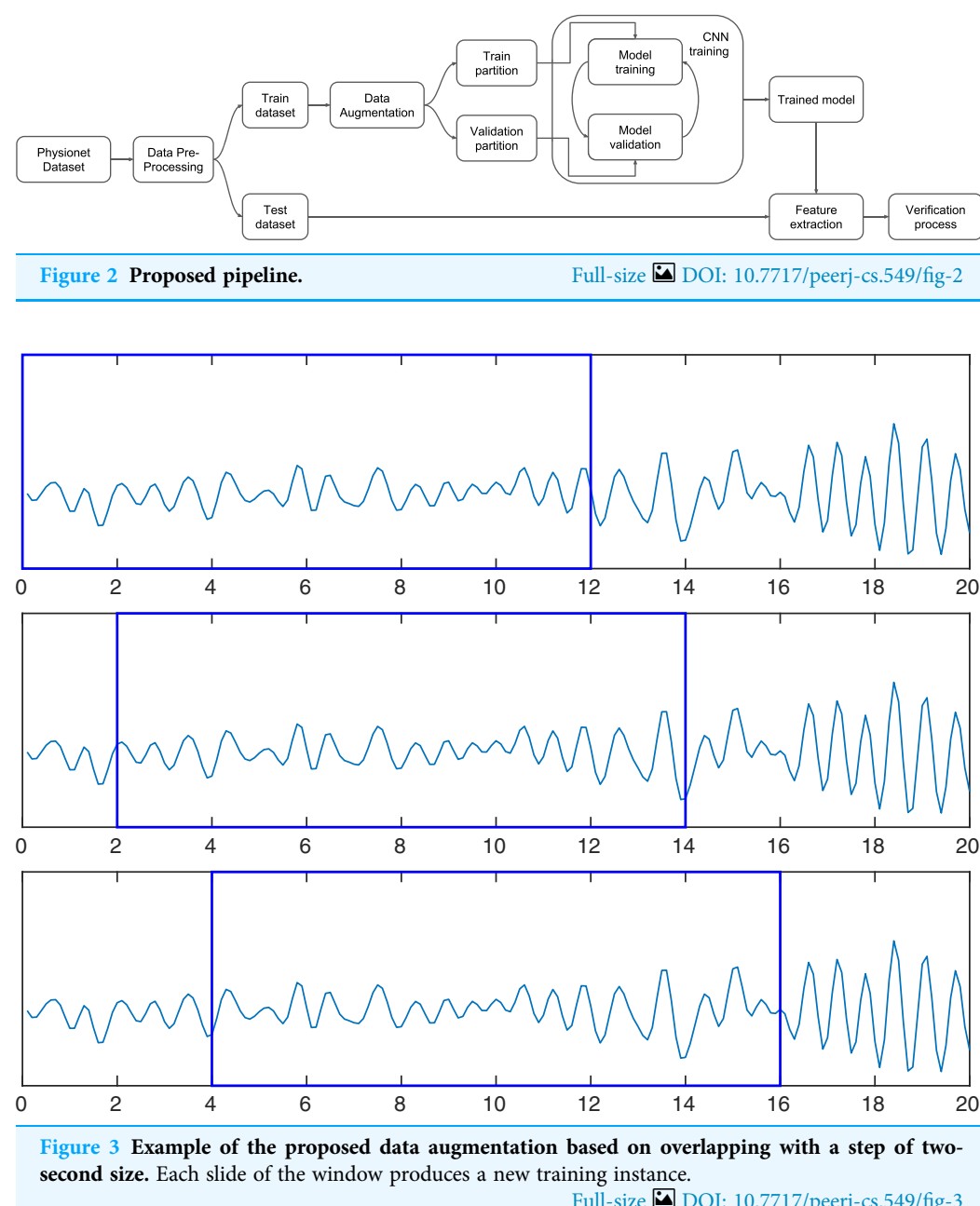

Figure 2 Proposed pipeline.           

**Figure 3 Example of the proposed data augmentation based on overlapping with a step of two-second size.** Each slide of the window produces a new training instance.

overlapping is investigated. The technique consists of creating new segments by a sliding window that moves by a constant step called stride. Figure 3 shows an example of the data augmentation technique used.

From the whole signal of an individual record, new segments are created. The extraction of the first segment starts from time zero and finishes within 12 s. The 12-second record represents 1,920 samples at a sampling frequency of 160 Hz. The next segment starts after a specific stride in *stride* seconds and finishes in *stride* + 12 s. The sliding window finishes at the end of the signal.

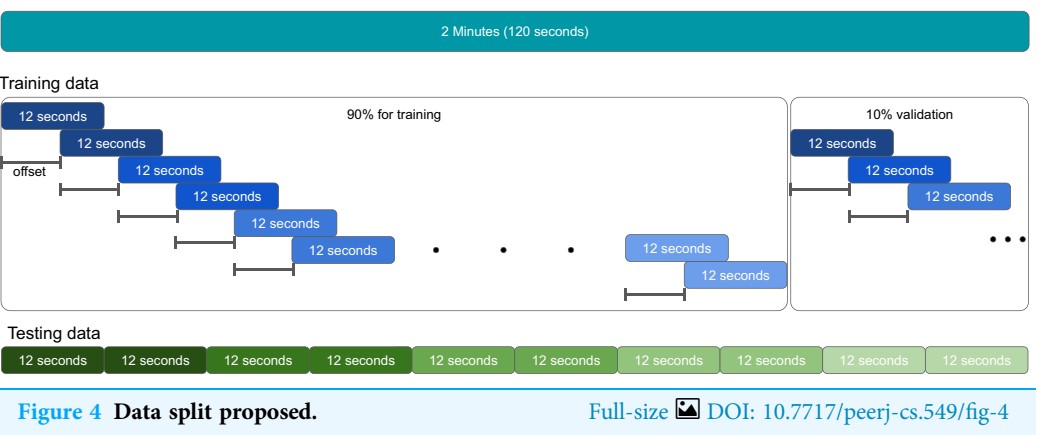

**Figure 4 Data split proposed.**

## CNN training and data representation

After the data augmentation step, the data is split into two subsets: train and validation. The first 90% signals resulted from the data augmentation are used for training, and the remaining 10% for validation (model adjusting during training), as presented in Fig. 4. It is worth highlighting that not necessarily 10% of the entire segment is used for validation, but 10% of the signals resulted from the data augmentation step. For example, if the Data Augmentation process output is 500 segments, the first 450 segments (90%) are for training and the last 50 segments (10%) for validation.

At first, the CNN architectures are evaluated with training and validation data. This evaluation is handled on the baseline evaluation perspective, once it is a more constrained scenario in comparison to the multi-task scenario perspective.

Two different one-dimensional CNN architectures with large receptive fields are investigated, once those with small kernels did not converge.

The two architectures are the result of the empirical adjustments in the depth and width according to the error measured in the validation data, during the training phase.

The first CNN architecture is composed of three convolution layers, each one followed by max-pooling, and four fully-connected layers with a dropout and a softmax layer. The convolution stride is set to one and no padding is used. The main among architectures investigated here is related to the number of convolutional layers. The second architecture is composed of five convolution layers with larger receptive fields in the initial layers. Both architectures are presented in Fig. 5.

Both architectures are evaluated according to the protocol proposed by *Fraschini et al. (2015)* and thus, the one that best fits the data can be determined.

Each architecture is trained for $n$ epochs, in which $n$ relies on the convergence of the training and validation error. The training process resembles a simple classification problem for an identification task (close-gallery) on a biometry context. Thus, a conventional supervised classification problem. Upon this fact, each sample provided to a CNN model is feed-forwarded and results in an output. This output is a probability that a sample belongs to one of the classes.

However, for the verification mode, one needs a feature vector as the output of a network instead of a probability vector of classes. To address this requirement, layers

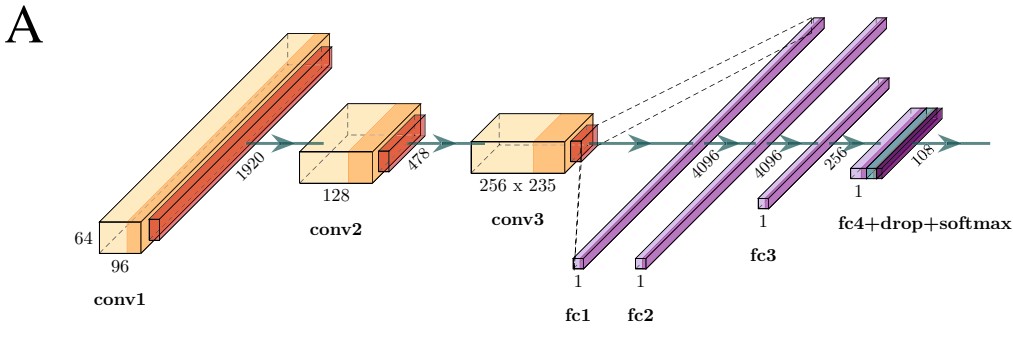

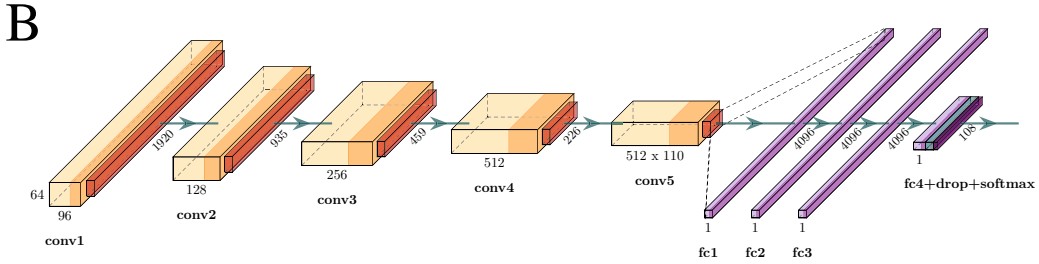

**Figure 5 Architecture evaluated in this work.** (A) Architecture proposed in which *conv1, conv2* and *conv3* have filters size equal to $11 \times 1$, $9 \times 1$ and $9 \times 1$, respectively, and stride equal to 1. *Pool1* with stride equal to 4 and *pool2* and *pool3* with stride equal to 2 and filter size equal to 2 for all three. The padding is equal to zero for all convolutional and pooling layers. (B) Architecture proposed in which *conv1, conv2, conv3, conv4* and *conv5* have filters size equal to $51 \times 1$, $17 \times 1$, $7 \times 1$, $7 \times 1$ and $7 \times 1$, respectively, and stride equal to 1 all five. *Pool1, pool2, pool3, pool4* and *pool5* are max pooling with filter size and stride equal to 2 for all pooling layers. The padding is equal to zero for all convolutional and pooling layers.

responsible for handling the classification (Soft-max, Dropout, and FC4. See in Fig. 5) should be removed. The feature extraction uses the FC3 output which can now be used as a feature vector.

From this point, the trained model starts to operate as a deep feature descriptor. Once a 12-seconds segment is provided to the network, the output is a feature vector of the same size as the FC3 output. For architecture A this size corresponds to 256 dimensions, while for architecture B, 4,096 dimensions.

## Biometric verification

Different from the training data in which data augmentation techniques were applied, there is no overlap for the testing data. That is, for the EC session, five segments of 12 s are extracted for each subject, while in the motor movement/imaginary tasks, ten segments, as shown in Fig. 4.

One of the most popular measures to compare methods for the verification mode in the literature is the Equal Error Rate (EER). This measure is the point where the False Acceptance Rate (FAR) and the False Rejection Rate (FRR) are equal in the Detection Error Trade-off (DET) curve. Comparing in an all-against-all scheme, a distance score (such as euclidean distance) is computed for each pair of feature vectors. Ideally, one

should have small scores between vectors from the same class (genuine) and large scores between vectors from different subjects (impostor).

In biometric verification mode, the number of impostor pairs is much higher than the genuine ones. A genuine pair occurs when two feature vectors are from the same individual. In contrast, an impostor pair corresponds to two feature vectors from two different individuals. As the number of impostor pairs is higher than the genuine ones, the biometric verification mode simulates a spoofing attack on a system. Thus, usually, the verification protocols is a challenging scenario.

## Evaluation scenarios

The first scenario adopted in this work is the one proposed by *Fraschini et al. (2015)* referenced in this work as baseline task evaluation. The evaluation protocol related to first scenario ensures the acquisition of the training data on the open-eye task, while the closed-eye task composes the testing data from 109 individuals. It is worth to highlight that the acquisition of training data and testing data came from two different sessions.

Therefore, the first studied scenario disregards the impact of the motor/imaginary tasks for the biometric verification. Contrasting to that, *Yang, Deravi & Hoque (2018)* carried out experiments to evaluate such conditions, i.e., the impact of motor activity on EGG signal for biometrics. Those experiments are the second scenario considered here and referenced as multi-task evaluation.

*Yang, Deravi & Hoque (2018)* proposed three experimental protocols to evaluate the EEG biometric in the EEG Motor Movement/Imagery dataset (*Goldberger et al., 2000*; *Schalk et al., 2004*). The protocols are:

1. Protocol P1: aims to investigate the influence of some regions of the scalp for biometric purposes. Three regions are selected: the frontal lobe (F), with electrodes AF3, AFz, and AF4, the motor cortex (M), with the electrodes C1, Cz, and C2, and the occipital lobe (O), with O1, Oz, and O2 electrodes. For each training, the signals from R1 and R3 of one task were selected and R2 from the same task was used for testing.

2. Protocol P2: explores the impact of training in the first and the third runs of the same motor/imaginary task and testing in the second run of all the remaining tasks and the baseline ones. Using only a subset of nine electrodes, 24 experiments were evaluated.

3. Protocol P3: aims to explore the impact on the combination of different motor and imaginary tasks for training. The authors reserve the second run of the first task (T1R2) to evaluate the performance of the training tasks. They start training with T1R1 and T1R3 separately and finish with the fusion of both with the other tasks and runs (all runs of T2, T3, T4, and T4).

The protocols P1 and P2 use the second run of all tasks as testing. In the protocol P3, only the task T1R2 is used as testing. Furthermore, all protocols are evaluated from both biometric identification and verification scenarios.

The use of several data acquisition sessions/runs may be unfeasible for real biometric systems once it is onerous for the user, and, typically, the enroll session occurs only on the

first time the user gets in touch with the system. Considering that, a third evaluation scenario is proposed in which a new protocol to validate the proposed approach is evaluated, using one task from a single run (session) for training referenced as Cross-task evaluation. Inspired by the third protocol (P3) proposed by *Yang, Deravi & Hoque (2018)*, the T1R2 is fixed as the testing data, and all the remaining tasks and runs are used for training. It is worth stressing that different from *Yang, Deravi & Hoque (2018)*, the remaining tasks and their respective runs are used separately for each training setup, instead of combining them. In that sense, train and testing in different task types should be a more challenging scenario when compared with a scenario that uses the same task in both training and test sets.

In all the three presented scenarios, samples from the same individual were present in both the training and testing sets. It is well known that such evaluation scheme tend to favor the classifiers. To avoid this issue and to assess the robustness of the proposed approach, a fourth protocol is proposed, forcing the division between training sets and test oriented to individuals. That is, the individual used to train the model is not used to test/evaluate the approach. This scenario was called cross-individual evaluation. Therefore, the Deep Learning models are trained with signals from the first 55 individuals from the Physionet database and tested with the last 54.

Due to the large number of possible combinations involving the 12 imaginary/motor tasks and the objective of this study, only the Eyes Open and Closed conditions were used to evaluate this fourth scenario.

Since the cross-individual evaluation is a more complex scenario and could benefit from a more robust model, the impact of using a state-of-art technique called Squeeze-and-Excitation Network (*Hu, Shen & Sun, 2018*) is evaluated. Section details the Squeeze-and-Excitation Networks.

### Squeeze-and-excitation network

The Squeeze-and-Excitation (SE) block is an architecture unit first introduced by *Hu, Shen & Sun (2018)* in order to capture the dependency between the convolution channels of a network. It has two main steps: Squeeze and Excitation.

The Squeeze step aggregates the global information of all the input channels and shrinking the spatial dimensions in a feature map space $U$ by using the global average pooling. The Excitation step transforms each channel of the feature map $U$ with weights obtained with the output of the previous step modulated for each channel.

One can easily obtain an SE network by stacking several SE blocks, replacing some components, or integrating SE blocks in some already known architecture, which is one of the great advantages of this method. In addition, in *Hu, Shen & Sun (2018)* the authors revealed state-of-the-art performance across multiple datasets and applications using models based on SE blocks.

Toward this direction, it is investigated the use of these units in the best model obtained in the other three scenarios evaluated. In order to determine the best SE network, the first step was to reproducing one experiment of the baseline task evaluation scenario (eyes open for training and eyes closed for testing) and trained a series of SE models using all 109

individuals for training. The performance of five models obtained by merging SE blocks with the best architecture are investigated as follows:

- SE Model 1: all convolution layers were replaced by SE blocks;
- SE Model 2: add a single SE block after the input layer;
- SE Model 3: add a single SE block before the fully connected layers, that is, after all convolution and pooling layers;
- SE Model 4: add a SE block after the input layer and before fully connected layers;
- SE Model 5: add a SE block after each convolution layer.

It is also evaluated the most promising reduction factor (hyper-parameter of SE blocks) and analyzed six different values for all models.

## RESULTS AND DISCUSSION

This section presents and discusses the protocols, the experimental setup and the obtained results.

### Experimental setup

All the CNN operations are conducted using the TensorFlow library, along with the high level API Keras. The specifications of the experimental computational environment are 64 GB of DDR4 RAM, an Intel (R) Core i7-5820K CPU 3.30 GHz 12-core, and a GeForce GTX TITAN X GPU. The source is available on GitHub (https://github.com/ufopcsilab/EEG-Multitask).

Two CNN architectures are evaluated with three band-pass filters for the baseline task: eyes open (EO) for training/validation and eyes closed (EC) for testing. First, the performance of the CNN architectures are evaluated on this simple task evaluation protocol by training for 20 epochs. Then, the effect of using different strides on the CNN performance is investigated which also tends to change the data volume created by the data augmentation technique and the time to train the CNN model. Thus, the best stride parameter is used for the remaining experiments.

Model weights are randomly initialized, and the optimization method used here is the Stochastic Gradient Descent with momentum coefficient of 0.9. A 10% dropout sub-sampling operation is placed before the softmax layer in order to minimize the over-fitting. Using a learning rate sequence of $lr = [0.01^2, 0.001^{18}]$, in which the number superscript represents the amount of epochs using that learning rate. For the present experimentation, there is no improvement when training for more than 20 epochs. After the training and the removal of the last layers of the CNN, the model resembles one embedding model or a deep feature descriptor for a 12 s length EEG input signal.

The data augmentation technique applied to the training data created a total of 384 samples for each individual from the EO data. For the motor (or imaginary) tasks (T1–T4), it was extracted from each individual a total of 889 or 905 samples (signals with 123 s and 125 respectively). In the verification mode, the data augmentation technique produces a total of 1,086 genuine pairs (intra-class) and 146,610 impostor pairs (inter-class) for

**Table 2 EER and decidability reported of the two proposed architectures. EER presented in percentage.**

| Frequency band | EER (%) | Decidability (%) | Architecture |
|---|---|---|---|
| 10–30 | 5.06 | 3.22 | |
| 30–50 | 0.19 | 7.02 | A |
| 01–50 | 9.73 | 2.50 | |
| 10–30 | 6.85 | 2.84 | |
| 30–50 | 0.65 | 3.61 | B |
| 01–50 | 9.64 | 2.20 | |

**Table 3 EER reported on EO-EC. EER presented in percentage. [#]Different evaluation protocol.**

| Reports | Approach | EER (%) |
|---|---|---|
| *Fraschini et al. (2015)* | Eigenvector Centrality | 4.40 |
| *Sun, Lo & Lo (2019)*[#] | CNN + LSTM | 0.41 |
| Proposed Method | CNN | 0.19 |

the evaluation on the baseline tasks (EO-EC). A total of 4,809 intra-class pairs and 571,392 inter-class pairs were generated for the multi-task (Motor-Imaginary) evaluation. In that sense, the multi-task evaluation is a more complex and computationally expensive than the simple tasks evaluation context.

## Baseline tasks evaluation

The results regarding the combination of CNN architectures and the frequencies-bands are presented in Table 2.

Architecture B, represented by Fig. 5B, reached an EER of 0.65% in 30–50 frequency band. Architecture A, depicted in Fig. 5A, achieved the best overall figures (0.19% EER) for the gamma frequency (30–50 Hz). The results in terms of decidability are also reported. The decidability is a measure of how separable two distributions are *Ratha, Senior & Bolle (2001)*. In this case, it indicates how far the impostor scores are from the genuine scores. As one may see in Table 2, architecture A performs better for all frequency bands, specially for the gamma frequency.

Both architectures A and B perform well for the 30–50 Hz band and worse for 01–50 Hz, but architecture A outperforms architecture B. One of the possible reasons is the filter sizes, which are smaller in the first one and could lead to more details by extracting local features.

In Table 3 is presented the EER from the proposed approach against the work proposed by *Fraschini et al. (2015)*. As one can see, the proposed method with the use of convolution neural networks significantly reduced the EER.

The experiments are under the same scenario: all electrodes (64), all individuals (109), training in EO (eyes open), and testing in EC (eyes closed). Furthermore, as the test protocol used is the same, a comparison becomes valid between both methods.

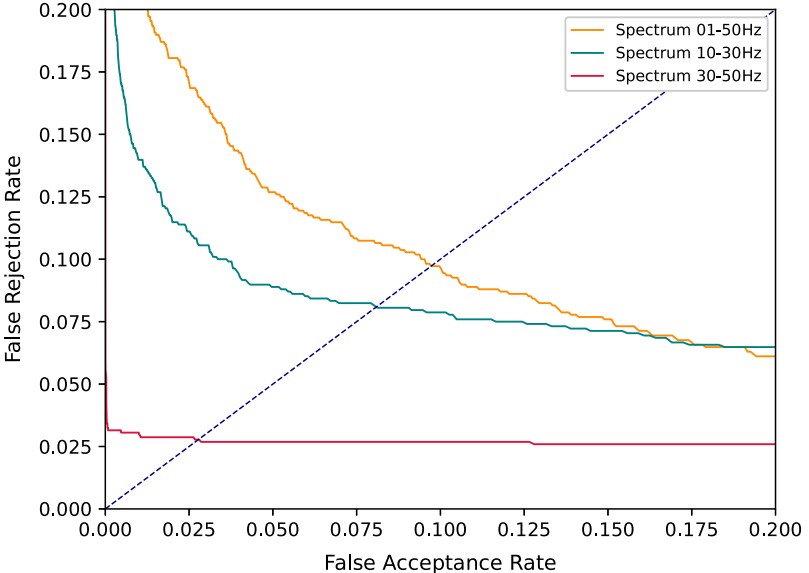

**Figure 6 DET curves comparing all spectrum evaluated with Architecture A.**

**Table 4 EER obtained for different strides. EER presented in percentage.**

| Stride | 20 | 40 | 60 | 80 | 100 | 120 | 140 | 160 | 180 | 200 |
|---|---|---|---|---|---|---|---|---|---|---|
| EER | 0.09 | 0.09 | 0.37 | 0.76 | 0.65 | 0.65 | 0.74 | 0.83 | 0.37 | 1.11 |

Despite the outstanding results reported by *Sun, Lo & Lo (2019)*, the evaluated scenario is different from the one proposed in *Fraschini et al. (2015)* and used here.

Figure 6 shows the performance of the architecture A employing the DET curve over all three frequency bands evaluated. The curve related to the spectrum of 30–50 Hz surpasses all other experiments described in the literature. Those results are in line with the ones presented in *Fraschini et al. (2015)* in which the 30–50 Hz range frequency overcomes the other frequencies for EEG biometrics-based. The reported results in this work differ from those reported by *Fraschini et al. (2015)* in the sense that there is a discrepancy in other frequency ranges, which are more significant here.

### Stride evaluation

To evaluate the impact of different strides in data augmentation, the first step started using a stride of 20, which represents 125 milliseconds (ms) step in the time domain, to generate the samples aiming the baseline task (EO). Then, the size of the step was successively increased by 20 up to 200 and trained Architecture A for each one. In that sense, the impact of overlap from 125 ms to 1,250 ms is analyzed. Baseline task EC, without overlapping, was reserved for evaluation (test) in all cases. Table 4 shows the EER for each stride.

One may see from Table 4 that the stride increase harms the proposed approach performance in the verification task. There is a trade-off on the stride size, the greater the

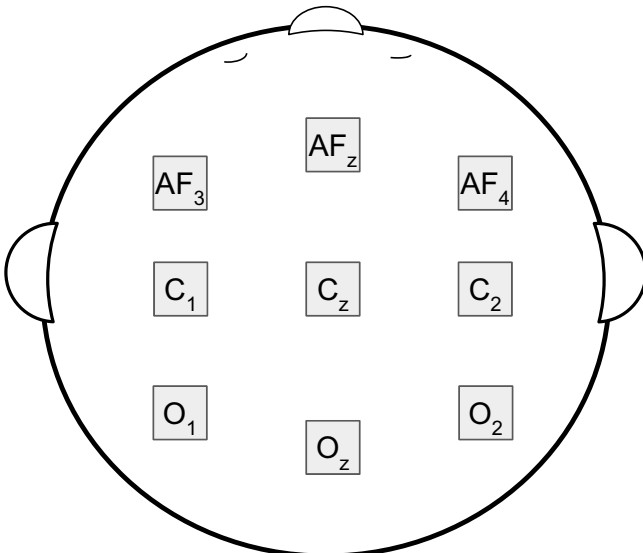

**Figure 7 Selected electrodes on motor cortex, frontal and occipital lobule.** Based on *Yang, Deravi & Hoque (2018)*.               

stride, the smaller are the number of samples generated and the greater is the EER. Upon this fact, the smallest stride (20) resulted in 38,208 training samples while the larger one (200) generated only 3,885. The larger stride obtained the worst result, with an EER of 1,11%. However, the stride of 40 produces the same performance of stride 20 and with half the number of samples for training. Thus, the stride equal to 40 is the one chosen to be used in the remaining protocols.

## Multi-task evaluation

The following experiments were based on the protocols proposed by *Yang, Deravi & Hoque (2018)*, which were presented in "Introduction". It was used the best CNN architecture and stride obtained on the above experiments to perform the experiments on the verification scenario.

### Protocol P1: region/task pairing

Figure 7 shows the location of the nine selected electrodes to perform this experiment.

The proposed method is evaluated with signals from the three regions of the scalp (see Fig. 7) combined. The results have shown that the performance is better using all nine electrodes. The best result is reported for T1R1 + T1R3 as training and T1R2 as test (EER of 0.12%). Among the experiments carried with only three electrodes, the best result is obtained with task four (T4) and motor cortex electrodes (EER of 0.85%). The worst results are related to the frontal lobe. A comparative analysis can be made with the results obtained in *Yang, Deravi & Hoque (2018)*, witch reached an EER of 7.83% when training and testing with the same task T1 using only the data from the occipital lobe. The same experiment performed here achieved 1.07% EER, showing that the proposed approach is robust even with a reduction in the number of channels.

**Table 5 EER reported of mismatch training/testing. EER presented in percentage.**

| Test | T1R2 | T2R2 | T3R2 | T4R2 | EO | EC |
|---|---|---|---|---|---|---|
| Train | | | | | | |
| T1R1+T1R3 | 0.12 | 0.29 | 0.42 | 0.42 | 0.10 | 0.37 |
| T2R1+T2R3 | 0.19 | 0.29 | 0.56 | 0.69 | 0.08 | 0.56 |
| T3R1+T3R3 | 0.21 | 0.19 | 0.29 | 0.19 | 0.18 | 0.36 |
| T4R1+T4R3 | 0.12 | 0,13 | 0.27 | 0.27 | 0.20 | 0.36 |

**Table 6 EER obtained by accumulating tasks/runs. EER presented in percentage.**

| Protocol | Train | Test | EER (%) |
|---|---|---|---|
| P3.1 | T1R1 | T1R2 | 0.10 |
| P3.2 | T1R3 | T1R2 | 0.44 |
| P3.3 | T1R1, T1R3 | T1R2 | 0.22 |
| P3.4 | T2R1, T2R2, T2R3 | T1R2 | 0.25 |
| P3.5 | T1R1, T1R3, T2R1 | T1R2 | 1.25 |
| P3.6 | T1R1, T2R1, T2R2, T2R3 | T1R2 | 1.14 |
| P3.7 | T1R1, T1R3, T2R1, T2R2 | T1R2 | 1.29 |
| P3.8 | T1R1, T1R3, T2R1, T2R2, T2R3 | T1R2 | 1.27 |
| P3.9 | T1R1, T1R3, T2R1, T2R2, T2R3, T3R1 | T1R2 | 1.58 |
| P3.10 | T1R1, T1R3, T2R1, T2R2, T2R3, T3R1, T4R1 | T1R2 | 1.47 |
| P3.11 | T1R1, T1R3, T2R1, T2R2, T2R3, T3R1, T4R1, T4R2 | T1R2 | 1.91 |

### Protocol P2: mismatch training/testing test

After applying the methodology described in "Introduction" with protocol P2, results showed no improvement when the train and the test come from the same task. The lowest EER, 0.08%, came from training with T2R1 + T2R3 and testing with the baseline task EO. Table 5 synthesizes the EER obtained for each experiment.

The proposed approach outperformed in all scenarios the best figures reported by *Yang, Deravi & Hoque (2018)* when training with T1 and testing with T2.

It is worth emphasizing that the physical (non-imaginary) tasks T3 and T4 are the most difficult for the biometric problem. Even when training and testing with tasks of the same nature (See Table 5), the results are worse. Our hypothesis is that in these cases, noise from muscle movement can interfere with signal acquisition.

### Protocol P3: heterogeneous training

In this protocol, the samples of the individuals are organized in mini-batches of size 100 and shuffled. All the nine electrodes from Fig. 7 are used and T1R2 fixed as test set for all experiments. Table 6 shows the tasks used for training and the EER obtained for each one.

As one may see on Table 6, stacking up different tasks hinder the proposed approach verification performance. The worst-case occurred in P3.11, which has a larger training set (EER of 1.91%). The best scenario occurs when the system is trained and evaluated

**Table 7 EER reported on T1R2. Both metrics are presented in percentages.**

| Reports | Train | Test | EER (%) |
|---|---|---|---|
| *Yang, Deravi & Hoque (2018)* | T1 & T2 | T1R2 | 2.63 |
| Proposed Approach | T1 & T2 | T1R2 | 0.27 |

with one run from the same task (EER of 0.1%). In some cases, accumulate tasks do not significantly affect performance.

### Comparative analysis

Protocol 1 led to the same conclusion then *Yang, Deravi & Hoque (2018)*: one could not determinate the most relevant electrode placement among the different regions of the scalp, but using all of them (all nine electrodes) enhances performance.

Protocol 2, both in this work and presented in *Yang, Deravi & Hoque (2018)*, showed that using different tasks to train and test does not affect performance. However, the proposed approach does not improve with the accumulation of tasks for training, different from reported by *Yang, Deravi & Hoque (2018)*. Moreover, a lower EER for all scenarios was reported, including Protocol 3.

One may see that the proposed CNN method has a significant improvement over the SOTA method proposed by *Yang, Deravi & Hoque (2018)*, considering that, the experiments were conducted under the same scenario with nine electrodes (See Table 7).

### Cross-task evaluation

To evaluate the cross-task scenario, a new protocol based in *Yang, Deravi & Hoque (2018)* is proposed. Among the 12 sessions, it is selected the T1R2 for testing. Besides, the remaining tasks (and runs) are used as training and validation. In that manner, eleven models to evaluate the T1R2 set are created. It is conducted the training following the methodology described in "Introduction". Also, the impact of using only nine electrodes (channels) instead of the 64 electrodes available on the Physionet Dataset is analyzed.

From the results presented in Table 8, one may see that CNN has converged for all training data, and the EER is lower than 0.39% for all of them. With 64 channels, the best scenario reaches 0.02% for T3R3 and an error, on average, of 0.16%. With nine channels, the lower EER achieved was 0.06% for T1R3 and the average error was 0.15%.

From these results, five questions arises:

*What is the impact of using different tasks from the same run in biometric verification?* Despite the parity of reported EER, the model trained with task T2 (same task as in T1, but in the imaginary state) presented a small and similar EER with both nine channels and 64, outperforming the models trained with tasks T3–T4. Despite task T3 showing the smallest EER with nine channels, it increased significantly with 64. In that sense, tasks from the same nature, even if it is imaginary, perform better than other ones. Although, the proposed CNN still performs well for the different tasks.

*What is the impact of using the same task but different runs in biometric verification?* From all experiments with 64 channels, the ones that use the same tasks from different runs had a good performance and a small variation. Upon this fact, one may affirm that

**Table 8 EER reported on T1R2. EER presented in percentage.**

| Test | Train | EER (%) 9 channels | EER (%) 64 channels |
|------|-------|--------------------|---------------------|
|      | T1R1  | 0.21 | 0.17 |
|      | T2R1  | 0.25 | 0.15 |
|      | T3R1  | 0.10 | 0.19 |
|      | T4R1  | 0.09 | 0.19 |
|      | T2R2  | 0.15 | 0.17 |
| T1R2 | T3R2  | 0.12 | 0.25 |
|      | T4R2  | 0.39 | 0.26 |
|      | T1R3  | 0.06 | 0.17 |
|      | T2R3  | 0.12 | 0.04 |
|      | T3R3  | 0.08 | 0.02 |
|      | T4R3  | 0.27 | 0.06 |

EEG biometry benefits of data acquisition in which the same task is performed. Using nine electrodes, the results presented high variance, which may happen due to the reduction in the number of channels used to train the model.

*What is the impact of using the same task but imaginary from different runs in biometric verification?* Different from the experiments using the T1 task, the experiments engaging the T2 task have a noticeable variation both with nine and 64 electrodes. Despite this fact, imaginary tasks perform similar to the ones which are not imaginary but with more variance. Toward this direction, EEG biometry benefits from data acquisition in which the task performed are the same in training and testing, even if it is imaginary or not.

*What is the impact of using different tasks (imaginary or not) in biometric verification?* Different tasks, motor or imaginary, for training and testing did not affect the performance of the system substantially. The biggest errors came from training with T4, a task completely different from the one used for test. On the other hand, training with a different motor task (T3) presented the same behavior from training with the same imagine task, aleatory, but satisfactory.

In that manner, EEG biometric may benefit from using tasks of the same nature, both motor tasks. In the same manner, it benefits from using the same task (imaginary or not). One possible hypothesis is that the activated areas from the brain are different from each task and the respective nature. The electrodes capture this behavior that directly influences the EEG curve. However, those changes are not substantial enough to affect very much the biometric verification process.

*What is the impact of the number of electrodes used in biometric verification?* Using fewer electrodes for training and testing did not affect considerably the performance of the system. Among all the performed experiments, the best result, 0.02% EER for training with T3R3, occurred with 64 channels, and the worst occurred with nine channels and an EER of 0.39%. This could indicate that EEG biometric benefits from using more channels, but the training accomplished with T3R1, T4R1, T2R2, T3R3, and T1R3, almost half of them, showed better performance with only nine.

**Table 9 Results obtained with SE Model 2 with r = 2 (SE2r2) and SE Model 3 with r = 32 (SE3r32) both trained with signs of half of the individuals.**

| Train | Test | CNN Arch. A | SE2r2 EER (%) | SE3r32 EER (%) |
|-------|------|-------------|----------------|-----------------|
| $EO_1$ | $EO_2$ | 0.55 | 0.18 | 0.92 |
|       | $EC_2$ | 0.93 | 0.51 | 0.39 |
| $EC_1$ | $EO_2$ | 0.40 | 0.41 | 0.55 |
|       | $EC_2$ | 0.97 | 0.36 | 1.11 |

One hypothesis to justify this result is that the use of 64 electrodes has redundancies and that using nine is enough to capture the most important signals emitted during the performance of a task, motor, or imaginary. Even in cases where the EER was higher using fewer channels, this increase was not significant. In that sense, one may conclude that using 64 channels adds unnecessary financial and computational cost since nine channels proved to be enough to achieve good results.

### Cross-individual evaluation

Since the Cross-individual evaluation is more challenging, it has more room for improvement. Thus, in this scenario, we evaluate the SE Blocks. We also compared networks based on SE blocks against networks based on simple CNN blocks under the same conditions.

In total, 30 models were evaluated on training data by changing the hyper-parameter $r$ six times for each of the five SE models proposed. The $r$ given by $r = 2^x$ ranged from 1 to 32 ($x = 0, 1, 2, 3, 4, 5$). The $r$ hyper-parameter is used to enhance the information of the channels **in the excitation operation** and it is related to the division of the original number of channels. If $r = 64$, the 64 original signals were divided 64 and the result will be only one channel in middle, which could mix up the original channels information.

The best results were achieved for SE Model 2 with $r = 2$, reaching an EER of 0.18% and even better for SE Model 3 with $r = 32$, with an EER of 0.11%, both greater than the results reported without the SE blocks. These two models are then used to evaluate the proposed Cross-individual scenario.

To refer to the first half of the dataset (55 first individuals), $EO_1$ and $EC_1$ are used. $EO_1$ is used to refer to the first half of the individuals present on the eyes open task and $EC_1$, for the first half of the eyes close task. $EO_2$ and $EC_2$ are the second half of the eyes open and eyes closed task respectively.

Table 9 shows the obtained results. One can observe that all reported results are closer to the reported in the baseline task evaluation (EER = 0,19%). This endorses the robustness of the proposed approach. SE Model 2 reached the best EER, 0.18% when testing with a different set of individuals under the same condition (eyes open). When trained with $EC_1$ the best result occurred for test with $E0_1$ (EER = 0.36%).

SE Model 3 showed lower performance than SE Model 2 for most experiments. Thus, we believe SE-block based network is a promising approach as a feature extractor for EEG signals.

## CONCLUSION

In the present work, it was evaluated the use of deep descriptors to extract features from the EEG signal for biometric purposes. The main focus was on one particular aspect: what is the impact of a motor (or even imaginary) activity in the EEG biometric modality? To investigate this question, the proposed approach with the Physionet EEG Motor Movement/Imagery Dataset was evaluated.

The proposed deep learning-based method achieved outstanding figures, overcoming the state-of-the-art methods even for the multi-task scenario. It is worth emphasizing that the proposed method is capable of performing well, even when trained on different tasks. For instance, it reaches an EER of 0.12% when trained with the task T4, corresponding to imagining moving the feet, and tested with the task T1, equivalent to performing the motor movement of closing the fists. One can conclude that the electrodes capture the motor (or imaginary) interference, however, those changes are not substantial enough to inhibit the biometric verification process. This work also evaluated the impact of the Squeeze-and-Excitation blocks for the EEG biometric problem. The Squeeze-and-Excitation blocks explore the interdependence between the channels of the signal, allowing better quality feature extraction and, therefore, better results.

Due to the nature of the proposed CNN-based approach, it allows simultaneous processing of all EEG channels, which is an advantage from a computational point of view. However, the use of multiple channels could turn the biometric modality impractical for real-world applications. Thus, the impact of using fewer channels (electrodes) is investigated, and a possible conclusion is that the use of all channels does not necessarily improve the result. The reported results showed that only nine electrodes allowed to achieve competitive results, or even better, compared to the use of 64 electrodes in similar tasks.

## ACKNOWLEDGEMENTS

We appreciate the reviewers and the editor for their detailed examination of our manuscript, which helped us to improve important aspects of the presentation.

### Funding

This work was supported by the Federal University of Ouro Preto (No. 5779), the National Council for Scientific and Technological (No. 313423/2017-2). The funders had no role in study design, data collection and analysis, decision to publish, or preparation of the manuscript.

### Grant Disclosures

The following grant information was disclosed by the authors:
Federal University of Ouro Preto: 5779.
National Council for Scientific and Technological: 313423/2017-2.

## Competing Interests

The authors declare that they have no competing interests.

## Author Contributions

- Mariana R.F. Mota performed the experiments, analyzed the data, performed the computation work, prepared figures and/or tables, and approved the final draft.
- Pedro H.L. Silva conceived and designed the experiments, performed the experiments, analyzed the data, performed the computation work, prepared figures and/or tables, and approved the final draft.
- Eduardo J.S. Luz conceived and designed the experiments, analyzed the data, authored or reviewed drafts of the paper, and approved the final draft.
- Gladston J.P. Moreira conceived and designed the experiments, analyzed the data, prepared figures and/or tables, authored or reviewed drafts of the paper, and approved the final draft.
- Thiago Schons conceived and designed the experiments, performed the computation work, prepared figures and/or tables, and approved the final draft.
- Lauro A.G. Moraes analyzed the data, authored or reviewed drafts of the paper, and approved the final draft.
- David Menotti analyzed the data, authored or reviewed drafts of the paper, and approved the final draft.

## Data Availability

EEG Motor Movement/Imagery raw data acquisition: https://physionet.org/content/eegmmidb/1.0.0/#files-panel.

The source is available on GitHub: https://github.com/ufopcsilab/EEG-Multitask.

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
