# Peer review of "A deep descriptor for cross-tasking EEG-based recognition"

_PeerJ Computer Science, doi:10.7717/peerj-cs.549_

## Round 0.1 · original submission · Major Revisions

The paper lacks technical details. More experiments are needed to establish the working of the proposed technique along with a comparative study of the competing solutions.

Reviewer 1 ·

Basic reporting

The topic of this paper is interesting and can be accepted in the PeerJ after do the following corrections:
1- What are the most challenging tasks to apply deep learning for EEG-based recognition?
this must be well addressed and mentioned in the Introduction section.
2- Section "PROPOSED APPROACH AND EVALUATION SCENARIOS" not explain well and needs to rewrite in a more scientific way. Also, there are some don't explain for example in Figure 3 "feature extraction" step but there aren't any details about it.
3- The authors used the EER measure to evaluate the proposed method. Did you try other measures?
4- What are the limitations of the proposed method?

Experimental design

1- More experiments are required.
2- The authors should use the newest EEG datasets.

Validity of the findings

1- More experiments with different validation measures are required.
2- The proposed method must be compared with the state of art to prove that its has a good performance.

Reviewer 2 ·

Basic reporting

1. Most contexts are overclaiming. For example, "We show that the proposed method is robust, even when trained and evaluated on different motor tasks
and fewer electrodes (nine channels). Due to deep learning hardware accelerators, we claim that our proposal is suitable to be embedded in a real-world application.".
Subjects performed different actions, but they are 1) motor-related tasks, 2) one single day with a one-time recording setup. I would recommend the author avoiding overclaiming.

2. This recent review article would help the author improve the Introduction on either general EEG applications or EEG-based biometrics https://ieeexplore.ieee.org/document/8945233.

3. One important work in the literature is missing. This is very recent work from late 2020. There are over 30 citations so far. https://ieeexplore.ieee.org/abstract/document/8745473. It is a deep learning approach for EEG-based biometrics.

4. Make the contribution list concise and address my comment in 1.

5. Figure 2 is not informative. The author can remove It.

6. Related works should be in a table form. The current form is complicated to read and to follow.

Experimental design

1. Remove "source: the author" from Figure 3, 5, ....

2. I am not agreed that data segmentation with overlapping is equal to data augmentation.

3. Validation is not explained clearly. Such as k fold cross-validation.

4. The author must compare their own proposed model to the previous works by reusing their code on the same data processing or by reproducing the code from their journal papers. It is not a fair comparison at this moment.

Validity of the findings

1. There is no statistical testing.

2. There is a slight novelty in using EEG from the different motor-related actions, which the author called multi-tasks.But, I would suggest the author make the deep learning model more novel. Or the author may include some more research questions to make this work solid. The novelty of the current form might not be enough.

Additional comments

Good luck with your chances on a revision.

Reviewer 3 ·

Basic reporting

The authors propose a biometric recognition system relying on EEG data, specifically addressing the issue of cross-task recognition.

This topic is indeed relevant yet not novel. The authors do not cite several recent relevant studies, such as
- Del Pozo-Banos et al, Evidence of a Task-Independent Neural Signature in the Spectral Shape of the Electroencephalogram, 2018;
- Vinothkumar et al, Task-Independent EEG based Subject Identification using Auditory Stimulus, 2018
- Kong et al, Task-Independent EEG Identification via Low-rank matrix decomposition, 2018;
- Fraschini et al, Robustness of functional connectivity metrics for EEG-based personal identification over task-induced intra-class and inter-class variations, 2019;-
- Kumar et al, Subspace techniques for task independent EEG person identification, 2019;

The novelty of the performed study with respect to the aforementioned works should have been deeply detailed. In more detail, several of the aforementioned studies use CNN to extract features from EEG, further affecting the contribution of the present manuscript.

Experimental design

There is a severe falw in the employed experimental design. The authors use CNNs to extract discriminative features from EEG data. The employed CNNs are trained over the same subjects then used for testing. Such conditions are impossible to be replicated in real life: they would imply any potential malicious subject to be available during the enrolment of a user. The obtained equal error rates are therefore achieved in unproper conditions. A proper verification test should be carried out over subjects which have not been considered during the training of the employed network,

Validity of the findings

A major issue of the paper regards the employed database. It is widely known that the PhysioNet database has been acquired performing a single recording session for each subject. Although several tasks are performed by each subject, they are carried out during the same session. The acquired data are extremely dependent on the session-specific conditions (for instance, the EEG recording device is never took off by the subjects during the whole session). the obtained fundings are therefore not reliable. It is recommended to perform tests on database collected, for each subject, during different days.

Additional comments

Given that CNNs have been already used for EEG-based biometric recognition (also considering cross-task conditions), a discussion about the novelty of the proposedapproach has to be given. Moreover, a performance comparison between the effectiveness of the proposed approach and the literature ones should be included.

---

## Round 0.2 · accepted · Accept

The authors have put substantial effort to rectify the issues identified and accommodate the changes suggested by the reviewers. The revised manuscript is in a good shape to be accepted for publication in PeerJ Computer Science.

Reviewer 2 ·

Basic reporting

no comment

Experimental design

no comment

Validity of the findings

no comment

Additional comments

Thank you for addressing all comments.